# Deficiency of Energy and Nutrient and Gender Differences among Chinese Adults: China Nutrition and Health Survey (2015–2017)

**DOI:** 10.3390/nu16142371

**Published:** 2024-07-22

**Authors:** Xiaoqi Wei, Liyun Zhao, Hongyun Fang, Mulei Chen, Wei Piao, Lahong Ju, Shuya Cai, Yuxiang Yang, Yuge Li, Fusheng Li, Jiaxi Li, Jing Nan, Dongmei Yu

**Affiliations:** 1Chinese Center for Disease Control and Prevention, NHC Key Laboratory of Public Nutrition and Health, National Institute for Nutrition and Health, Beijing 100050, China; weixq@ninh.chinacdc.cn (X.W.); zhaoly@ninh.chinacdc.cn (L.Z.); fanghy@ninh.chinacdc.cn (H.F.); piaowei@ninh.chinacdc.cn (W.P.); julh@ninh.chinacdc.cn (L.J.); caisy@ninh.chinacdc.cn (S.C.); yangyuxiang1996@sina.com (Y.Y.); liyuge1122@163.com (Y.L.); 18246668833@163.com (F.L.); jiaxijy6@163.com (J.L.); nj13939012762@163.com (J.N.); 2Chinese Center for Disease Control and Prevention, Beijing 102206, China; chenml@chinacdc.cn

**Keywords:** micronutrient inadequacies, BMI, adults, China

## Abstract

Half of Chinese adults face the double burden of overweight/obesity and micronutrient deficiencies, and nearly 40% of them are severely overweight/obese or have micronutrient deficiencies. This study used the data from China Nutrition and Health Survey (CNHS) from 2015 to 2017 to estimate the prevalence of inadequate dietary micronutrient intake (including vitamin A, vitamin B_1_, vitamin B_2_, vitamin C, cCalcium, iron and sodium) in Chinese adults and further determine the differences in micronutrient intake by gender, age and BMI. A total of 61,768 subjects were included in this study, of which 33,262 (54%) were female. The intake of energy and all macronutrients decreased with age, and the intake was higher in men than in women. Inadequate energy intake occurs in adults of all ages. In terms of nutrient intake, women had a higher rate of insufficient carbohydrate intake than men in all age groups. Inadequate protein intake was more common in women aged 18–49 years (60.9%) than in men. Compared with women, men had a higher rate of vitamin B2 intake. Insufficient vitamin B3 intake was more common in women aged 18–49 years (35.6%), men aged 65–79 years (39.7%) and men aged 80 years and above (47.9%). In all age groups, insufficient vitamin C intake is higher in women than in men—up to 85.8 percent in women aged 80 years old and above. Compared with men in the same age group, insufficient intake of calcium and iron is more obvious in women. Women have significantly higher rates of inadequate intake of calcium, iron and sodium than men. In the analysis of correlations between BMI or demographic data and micronutrient intakes, the likelihood of micronutrient intakes being insufficient was higher in the central and western regions in all age groups compared to the eastern regions. The risk of insufficient micronutrient intake was higher in obese men and women aged 18–49 years and 50–64 years. Underweight and overweight women in the 65–79 age group were more likely to have inadequate micronutrient intake. Obese women over 80 years of age were less likely to have inadequate micronutrient intake. No significant difference was found between urban and rural areas for each age group.

## 1. Introduction

The amount of micronutrients (including vitamins and minerals) required by the human body is relatively small (measured in milligrams or micrograms), but they are widely involved in metabolism and physiological functions and play an important role in growth and the development and maintenance of body immunity. Micronutrient borderline deficiencies (also known as subclinical deficiencies) can increase the risk of certain degenerative diseases. In the case of B vitamins (VB), deficiency of VB2 can increase the risk of colon cancer [1]. The proportion of micronutrient margin deficiencies in the population is often much higher than the deficiency rate; for example, only 5% of Canadian adults aged 19 years and older have a VB12 deficiency, while the margin deficiency rate is as high as 19% [2]. In addition, excessive amounts of some micronutrients can also cause corresponding health damage, and excessive amounts of niacin can cause serious adverse reactions, including glucose intolerance, insulin resistance and liver damage [3]. Studies have shown that type 2 diabetes (T2DM) and childhood obesity may be related to excessive niacin intake [4], and excessive iron is a risk factor for T2DM patients, affecting most of their basic functions to reduce insulin secretion and insulin resistance [5].

In people with chronically high intakes of B vitamins and iron, there is a high risk of impaired glucose tolerance, insulin resistance and subsequent obesity and T2DM. The Report on Nutrition and Chronic Diseases of Chinese Residents (2020) [6] points out that some prominent nutritional problems of Chinese residents are mainly reflected in the following three aspects: First, the problem of unreasonable dietary structure is prominent, the dietary fat supply ratio continues to rise, the intake of edible oil and edible salt is far higher than the recommended value, and the consumption of fruits, beans, soybean products and milk is insufficient. Second, the situation of overweight and obesity in China is grim, and the rate of overweight and obesity in urban and rural residents of all ages continues to rise. Third, the deficiency of important micronutrients in some key areas and key groups, such as infants, women of childbearing age and the elderly, still needs attention.

The National Nutrition Plan (2017–2030) [7] (hereinafter referred to as the Plan) issued and implemented by the General Office of the State Council in 2017 clearly put forward phased targets for micronutrient deficiency, including the control of iron deficiency anemia rate, folic acid deficiency rate and the strengthening of iodine nutrition monitoring. In China, 44% of adults face the burden of overweight/obese or micronutrient deficiencies, of which nearly 40% of people have both overweight/obesity and micronutrient deficiency of common retinol, thiamine, riboflavin, vitamin C, calcium, selenium, zinc, magnesium or other key micronutrients. More than 80% of adults have at least two vitamin or mineral deficiencies [8]. The rate of overweight and obese adults in China is 65.3%, and the number of overweight and obese people may reach 790 million. Obesity will also impose a burden on China’s health care system. By 2030, it is conservatively predicted that the medical expenses caused by obesity in China may account for 22% of the total medical expenses in the country [9]. In previous studies, women had lower intakes of vitamins A and C, thiamine, riboflavin, calcium, phosphorus, calcium, phosphorus, iron, zinc, copper and selenium than men. However, there are few studies on the comprehensive evaluation of micronutrient intake in Chinese adults, especially on the comprehensive evaluation of underweight/overweight/obesity and micronutrient deficiency in the same individual and on the gender classification of adults. Therefore, exploration is needed to address this limitation and clarify whether daily dietary micronutrient status varies by BMI category.

In this paper, using data from the China Nutrition and Health Survey (CNHS) from 2015 to 2017, we estimate the deficiency rate of comprehensive micronutrient intake in the diets of Chinese adults. The differences in micronutrient intake by sex, age and BMI were further determined. These findings will provide an important scientific basis for policy makers to implement interventions to improve undernutrition and overnutrition in China.

## 2. Materials and Methods

### 2.1. Study Design and Samples

The data in this study were collected from the China Nutrition and Health Survey (CNHS) from 2015 to 2017. The 2015 China Adults Chroni Diseases and Nutrition Survey (CACDNS 2015) [10,11] adopted the method of multi-stage cluster random sampling, and the survey subjects were adults aged 18 years and above. Chinese adult chronic disease and nutrition surveying was conducted at 302 survey sites in 31 provinces across the country (Table A1). At each surveying site, a multi-stage stratified cluster random sampling method was used to select the participants. Three townships were randomly selected from each monitoring point, and two administrative villages were randomly selected from each township. In each selected administrative village, households were divided into several villager/resident groups with a scale of not fewer than 60 households, and one villager/resident group was selected by simple random sampling. In each sampled villager/resident group, 45 permanent residents aged 18 and above were selected to conduct a personal questionnaire survey on chronic diseases and nutrition. After data cleaning, a total of 61,768 adults aged 18 and above participated in this study (Table A2).

### 2.2. Data Collection and Measurements

The questionnaire designed by the national project team was adopted, and the information from the participants was collected by the uniformly trained local CDC staff through face-to-face inquiry, including the basic registration form of the household and family members.

Dietary survey information was derived from the registration form regarding the number of times of instances of family cooking and dining over 3 days and 24-h dietary recalls. The household consumption of edible oil, salt, monosodium glutamate and other major condiments for 3 consecutive days (two weekdays and one weekend) was investigated by weighed record, and 24-h dietary recalls were used to collect all the food, including staple food, non-staple food, snacks, fruits and drinks, eaten at home and out in 24 h for 3 consecutive days (two weekdays and one weekend). The consumption of household edible oil and condiments during the survey period was recorded by the weighing accounting method, and the intake of household edible oil and condiments was divided according to the number of family meals and the proportion of personal energy of family members according to personal intake. Personal food consumption includes the consumption of food, oils and condiments in a 24-h dietary recall. The per capita daily dietary intake of energy, vitamins and minerals in the average daily diet of each person was calculated according to the 2004 [12] and 2009 [13] editions of China Food Composition Table. The nutrients in the food composition list consist of 10 kinds of energy, water, ash, dietary fiber and macronutrients, 15 kinds of vitamins, 11 kinds of minerals, 20 kinds of amino acids and 38 kinds of fatty acids.

Height and weight were measured centrally by uniformly trained investigators using a certified measurement instrument designated by the National Project Team (TZG type height and sitting altimeter and Tanita HD-390 electronic weight scale). The accuracy of weight scale and height sitting altimeter are 0.1 cm and 0.1 kg, respectively. When measuring height and weight, subjects were required to take off their shoes, hats and thick clothes.

### 2.3. Data Processing and Data Analysis

(1) Energy and nutrients were calculated according to the China Food Composition Table from 2004 [12] and 2009 [13], which describes the daily intake of energy and various nutrients in the average person’s diet. The data were cleaned to remove people with energy intakes of less than 800 kcal or more than 5000 kcal per standard person per day or less than 1 day in the 3-day energy intake survey. The dietary reference intakes (DRIs) of Chinese residents [14] were used to analyze the differences in dietary intake, total energy intake and micro/macro nutrient intake between different genders (Table A3). The DRI includes the following components: energy requirement (EER), acceptable macronutrient distribution range (AMDR), mean requirement (EAR), adequate intake (AI) and tolerable maximum intake (UL). The dietary reference intakes (DRIs) of Chinese residents [14] were used to analyze the differences in dietary intake, total energy intake and micro/macronutrient intake between different genders. The DRI includes the following components: energy requirement (EER), acceptable macronutrient distribution range (AMDR), mean requirement (EAR), adequate intake (AI) and tolerable maximum intake (UL).

(2) Body mass index (BMI) was calculated with the following formula: body weight (Kg)/height2(m^2^); the cutoff of BMI categories was based on the Criteria of Weight for Adults according to the health industry standard of China, WS/T 428-2013: underweight: BMI < 18.5 kg/m^2^, normal: 18.5 kg/m^2^ ≤ BMI < 24 kg/m^2^, overweight: 24 kg/m^2^ ≤ BMI < 28 kg/m^2^ and obese: BMI ≥ 28 kg/m^2^.

(3) Inadequate micronutrient adequacy: inadequate micronutrient adequacy as <4 adequate micronutrient intakes (out of 8 in total, including vitamin A, vitamin B_1_, vitamin B_2_, vitamin C, calcium, iron and sodium).

(4) Sociodemographic indicators: (i.) gender is divided into male and female; (ii.) the age groups were divided into 18 to 49 years old, 50 to 64 years old, 65 to 79 years old and 80 years old and above; (iii.) the region is divided into eastern, central and western regions; (iiii.) areas are divided into urban and rural areas.

### 2.4. Statistical Analysis

SAS software (version 9.4, SAS Institute Inc., Cary, NC, USA) was used for data cleaning and calculation. For statistical descriptions, percentage (%) was used for categorical variables, and mean and standard deviation (±SD) were used for continuous variables. Nutrient adequacy was assessed using the form of mean and standard deviation (±SD) when the distribution was normal, whereas the median and interquartile range (IQR) were used when the distribution was non-normal. For statistical analysis, chi-square tests were used to determine differences between groups, and to assess the association between nutritional status and demographic characteristics with inadequate and excessive nutrient intake, multivariate logistic regression analysis was performed. Statistical tests were performed for each age group and stratified by sex. *p*-values < 0.05 were considered statistically significant.

## 3. Results

### 3.1. Characteristics of Participants

A total of 61,768 participants were included in this study, among which 33,262 (54%) were female; within each age and sex group, the range of ages was almost the same. Most of them lived in eastern and rural areas. More than 50% of adults were overweight or obese. Significant gender differences were found in men aged 18–49 years, 50–64 years and 65–80 years, among which the rates of underweight and obesity in the 50–64 and 65–80 years were lower than those in the same age group (*p* < 0.01). Intakes of energy and all macronutrients decrease with age, and intakes are higher for men than women, as shown in Table 1.

### 3.2. Energy Intake Results

In total, 60.5%, 57.8%, 57.3% and 67.0% of adults aged 18–49, 50–64, 65–79 and 80 and above were found to have insufficient energy intake, respectively. Significant gender differences were found across all age groups, as shown in Table 2.

### 3.3. Macro- and Micronutrient Intake

Almost all of the survey participants had a severely inadequate intake of dietary fiber, while more than half had inadequate intakes of vitamins A, B1, B2, C, B1 and calcium. Insufficient iron intake was mainly concentrated in women aged 18–50 years (35.5%) and women aged 80 years and older (10.3%). Significant excess sodium intake was found in all age groups and in both men and women (Figure 1 and Figure 2).

### 3.4. Associations between Micronutrient Intakes and Gender

In all age groups, women had higher rates of insufficient carbohydrate intake than men. Among them, women aged 80 and older had a carbohydrate deficiency rate of 10.3%. Insufficient protein intake was more common in women aged 18–49 years (60.9%) than in men (*p* < 0.01), and insufficient fat and dietary fiber intake was more common in women aged 18–49 years, 50–64 years and 65–79 years. Insufficient vitamin A intake was more common in men aged 18–49 years (78.7%), 50–64 years (77.4%) and 65–79 years (77.7%) than in women. Vitamin B1 intakes were more common in women aged 18–49 years (83.1%) and men aged 65–79 years (90.1%). Compared with women, men aged 18 to 49 years (91.8%), 50 to 64 years (93.0%) and 65 to 79 years (94.8%) had a higher rate of insufficient vitamin B2 intake. Insufficient vitamin B3 intake was more common in women aged 18–49 years (35.6%), men aged 65–79 years (39.7%) and men aged 80 years and above (47.9%). In all age groups, insufficient vitamin C intake was higher in women than in men and up to 85.8 percent in women aged 80 years old and above. Compared with men in the same age group, insufficient intake of calcium and iron was more obvious in women, as shown in Table 3.

### 3.5. Association between Demographic Features, Nutritional Status and Inadequate Micronutrient

Significant associations were found between BMI or demographic data and micronutrient intakes in different age and sex groups, with a higher likelihood of micronutrient deficiency in central and western regions compared to eastern regions in all age groups. The risk of micronutrient deficiency was higher in obese men and women aged 18–49 years and 50–64 years. Underweight and overweight women in the 65–79 age group were more likely to have inadequate micronutrient intake. Obese women over 80 years of age were less likely to have inadequate micronutrient intake. No significant difference was found between urban and rural areas for each age group, as shown in Table 4.

## 4. Discussion

Micronutrients such as selenium, fluoride, zinc, iron and manganese are minerals that are essential for many homeostasis processes. The importance of these micronutrients begins early in the human life cycle and continues throughout its various stages. The intake of micronutrients (minerals and vitamins) has some health benefits, which are attributed to their role as co-factors and cofactors in enzyme systems, the formation of healthy bones and teeth, the maintenance of body tissues and other physiological and biochemical functions [15,16].

The aim of this study was to describe the impact of characteristics of sex, age and region on micronutrient intake among adults aged 18 years and older in China, with particular attention to possible gender differences, and to optimize current and design new nutritional interventions to improve the dietary intake of Chinese residents. Ideally, a balanced dietary pattern is the basis for maximizing human application needs and health. A diverse diet should include a rich variety of fruits and vegetables (vitamins A and C), whole grains (B vitamins), nuts (unsaturated fatty acids), oils (vitamin E), meat (protein) and legumes (zinc), dairy products (calcium) and aquatic products (vitamin D) to provide for daily nutritional needs [17]. The most obvious difference in this article is between the genders. In all age groups, women had higher rates of insufficient vitamin C intake than men. Compared with men in the same age group, women’s intakes of protein, calcium and iron were more pronounced. The most obvious age difference is that the intake of energy and all macronutrients decreased with age, and the intake was higher in men than in women. The proportion of insufficient energy intake was higher, and significant differences were found in the following age groups: 18–49 years, 65–79 years and 80 years and above. The most significant regional differences were the higher likelihood of micronutrient deficiency in the central and western regions in all age groups compared to the eastern regions. No significant differences were found between urban and rural areas, unlike in some studies, which showed that micronutrient deficiencies were more likely to occur in urban areas than in rural areas, according to a study of women of childbearing age in Malawi [18].

These findings are consistent with studies from different parts of the world [19,20] that describe gender differences in diet and nutrition. Studies conducted in developing, low-income countries have shown that compared with men, women have a lower intake of foods such as meat, eggs, milk, legumes, fruits and vegetables and a higher risk of micronutrient and energy deficiencies [21]. However, studies in Western developed countries [22] show that women tend to make healthier food choices and consume more fruits, vegetables and dietary fiber than men, which is related to socioeconomic changes, education level and dietary lifestyle.

We also found that obese men and women aged 18–49 years and 50–64 years were more likely to have insufficient micronutrient intake. The risk of micronutrient intakes was higher in obese men and women aged 18–49 years and 50–64 years. Overweight and obesity are very common around the world and can adversely affect an individual’s nutritional status. Studies have shown that many obese patients do not have sufficient intakes of minerals such as iron and calcium, as well as vitamins A and B12, which can be the result of poor diet quality. Being overweight or obese increases the risk of various chronic diseases affected by nutrition, including osteoarthritis, type 2 diabetes, cardiovascular disease, respiratory problems and cancer [23]. Regarding energy intake, we found that more than 50% of the participants in our study had an energy intake lower than the energy requirement for light physical activity (EER) (Table A3), and the majority of the population had normal BMIs, rather than overweight and obese. However, in most studies, insufficient energy intake was common in the 3-day 24-h survey, and the intake obtained according to the survey was 15% to 30% lower than the actual intake [24,25].

In general, unbalanced energy intake is often the result of unbalanced diet structure, and the per capita carbohydrate and protein intake in this study was lower than the recommended amount of dietary guidelines, while the fat intake was much higher than the recommended nutrient intake. Therefore, it is necessary to strengthen the publicity of reasonable diet and advocate a balanced diet pattern. The double burden of malnutrition (DBM), defined by the World Health Organization as a phenomenon in which “malnutrition or micronutrient deficiencies coexist with overweight, obesity, or noncommunicable diseases associated with malnutrition” [26], is a public health challenge in low- and middle-income countries. The monitoring of nutrition and the health status of Chinese residents from 2010 to 2012 [27] showed that intakes of vitamin A (<EAR: 77%), vitamin B1 (<EAR: 77.8%), vitamin B2 (<EAR: 90.2%), iron (<EAR: 11.5%) and calcium (<EAR: 96.6%) were insufficient. Clinical evidence shows that insufficient intake of certain micronutrients such as vitamin A, vitamin B1 or vitamin C is associated with obesity and metabolic syndrome [28]. Our study found that adult Chinese men and women have a high prevalence of obesity, accompanied by insufficient intakes of vitamins A, B1 and C, especially in adult women, possibly due to micronutrient deficiencies caused by unbalanced diets [29]. Based on this information, further research and intervention studies need to be considered, with a focus on undernutrition and overnutrition at all life stages in both men and women.

In our study, we found that excessive sodium intake was more common than insufficient sodium intake. High-sodium diets have become one of the major risk factors for stroke-related death and disability in China [30]. According to the Report on Nutrition and Chronic Diseases in Chinese Residents (2020), the prevalence of hypertension among adults aged 18 and above in China is 27.5% [6].The prevalence of hypertension in adults in the United States is approximately 44% to 49% [31]. On 19 September 2023, the World Health Organization released the first-ever report on the devastating global impact of high blood pressure, which states that high blood pressure affects one in three adults worldwide, about four out of five people with hypertension are undertreated, and that if countries can expand treatment coverage, 76 million deaths could be avoided between 2023 and 2050 [32].

There are three limitations in this study. Firstly, 24-h dietary review surveys rely on the memory of survey subjects, which can lead to recall bias, often resulting in an underestimation of actual intake. [33]. Secondly, the condiments such as oil and salt involved in this paper are consumed within the family home and do not include the condiments in eating out. Thirdly, in addition, the on-site surveys at all the survey sites in this study were not conducted at the same time, which caused certain errors in the calculation of seasonal food. However, this study was large, and the overall food consumption could be obtained quite reliably when the diet was relatively monotonous. Moreover, the survey time of the 24-h dietary review method was generally 3 consecutive days, including 2 working days and 1 rest day. A 24-h household survey was conducted every day. Although this survey was not carried out at the same time, during the calculation, repeated use on weekdays had reduced the deviation of food intake caused by seasonal and holiday factors. In further studies, personal information such as education and physical activity should be considered to determine if any of these are associated with health outcomes related to dietary intake.

## 5. Conclusions

The current findings suggest that nutrient intake and quality is inadequate for a majority of adults aged 18 years and older in China, confirming the problem of macronutrient imbalance and double burden malnutrition. The study also demonstrated an association between BMI or demographic information and multiple micronutrient deficiencies in different age groups. Understanding and controlling micronutrient deficiencies in Chinese adults may improve public health. Therefore, there is a need to strengthen nutrient supplements or dietary interventions for special groups such as women aged 50 years old and above and the elderly to address the current problem of micronutrient intakes in China through rational dietary promotion and encouragement of diverse and balanced dietary patterns.

## Figures and Tables

**Figure 1 nutrients-16-02371-f001:**
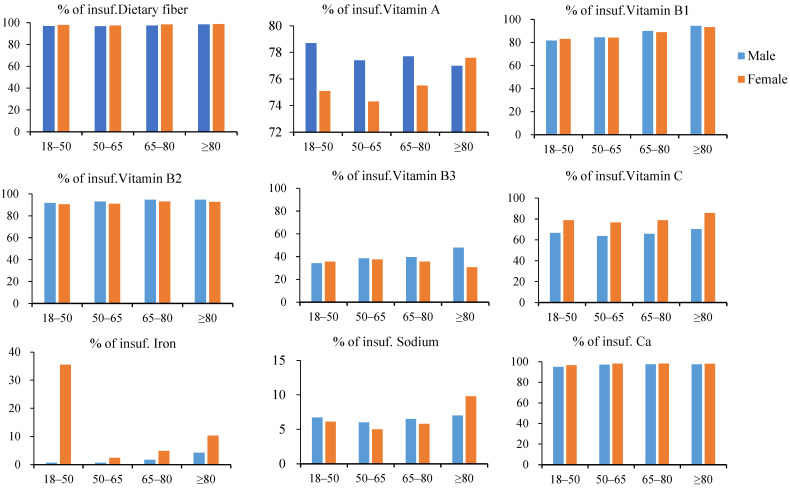
Proportions (%) of insufficient intake per age group and gender.

**Figure 2 nutrients-16-02371-f002:**
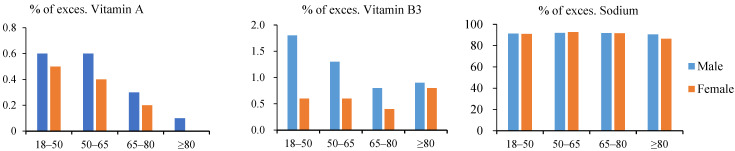
Proportions (%) of excessive nutrients per age group and gender.

**Table 1 nutrients-16-02371-t001:** Characteristics of demographics, BMI categories and dietary intake among Chinese adults in study population.

	18–49 y (*n* = 25,295)		50–64 y (*n* = 23,697)		65–79 y (*n* = 11,509)		≥80 y (*n* = 1267)	
	Male (*n* = 11,035)	Female (*n* = 14,260)	Male (*n* = 10,864)	Female (*n* = 12,833)	Male (*n* = 5933)	Female (*n* = 5576)	Male (*n* = 674)	Female (*n* = 593)
Demographics								
Region; *n* (%)								
Eastern	4197 (38.0)	5408 (37.9)	4398 (40.5)	5256 (41.0)	2364 (39.8)	2224 (39.9)	310 (46.0)	288 (48.6)
Central	2967 (26.9)	3959 (27.8)	3272 (30.1)	3890 (30.3)	1854 (31.2)	1673 (30.0)	184 (27.3)	161 (27.2)
Western	3871 (35.1)	4893 (34.3)	3194 (29.4)	3687 (28.7)	1715 (28.9)	1679 (30.1)	180 (26.7)	144 (24.3)
Urban/rural; *n* (%)								
Urban	4389 (39.8)	5827 (40.9)	4366 (40.2)	5297 (41.3)	2611 (44.0)	2591 (46.5)	310 (46.0)	277 (46.7)
rural	6646 (60.2)	8433 (59.1)	6498 (59.8)	7536 (58.7)	3322 (56.0)	2985 (53.5)	364 (54.0)	316 (53.3)
Anthropometrics								
Weight (kg); mean (SD)	68.9 (12.1)	58.8 (10.0)	66.7 (11.0)	59.2 (10.1)	63.3 (11.0)	56.1 (10.3)	60.2 (11.1)	51.3 (10.5)
Height (cm); mean (SD)	168.2 (6.6)	156.9 (5.9)	165.6 (6.5)	154.6 (6.0)	163.2 (6.6)	151.7 (6.4)	161.1 (6.9)	148.4 (7.3)
BMI categories; *n* (%)								
*p*-value ^a^	<0.0001		<0.0001		<0.0001		0.2995	
Underweight	400 (3.6)	632 (4.4)	267 (2.5)	379 (3.0)	303 (5.1)	293 (5.3)	50 (7.4)	48 (8.1)
Normal	5068 (45.9)	7385 (51.8)	5038 (46.4)	5301 (41.3)	2970 (50.1)	2460 (44.1)	368 (54.6)	335 (56.5)
Overweight	3891 (35.3)	4433 (31.1)	4129 (38.0)	4957 (38.6)	2054 (34.6)	1970 (35.3)	201 (29.8)	151 (25.5)
Obese	1676 (15.2)	1810 (12.7)	1430 (13.2)	2196 (17.1)	606 (10.2)	853 (15.3)	55 (8.2)	59 (9.9)
Dietary intake; median (IQR)								
Energy (kcal)	2060.8 (808.3)	1690.1 (652.7)	1975.5 (783.6)	1655.7 (633.8)	1809.1 (717.5)	1524.5 (589.9)	1608.3 (612.3)	1380.3 (555.6)
Macronutrients (g)								
Carbohydrates	259.3 (121.3)	221.9 (99.5)	256.5 (120.3)	225.6 (98.9)	242.8 (110.1)	211.2 (94.3)	216 (93.8)	188.5 (87.5)
Protein	61.6 (29.8)	50 (24.0)	58 (27.6)	48.2 (23.6)	52.6 (25.5)	44.3 (22.1)	47.3 (22.7)	39.5 (21.6)
Fat	80.1 (51.9)	63.7 (40.6)	74 (48.8)	60 (39.7)	64.7 (43.3)	52.7 (35.3)	57.7 (38.3)	49.9 (34.1)
Dietary fiber	8.9 (6.0)	8.1 (5.3)	9 (6.1)	8.2 (5.5)	8.4 (5.8)	7.6 (5.2)	7.4 (5.4)	6.2 (4.0)

^a^ *p*-values are based on chi-square tests

**Table 2 nutrients-16-02371-t002:** Proportion of insufficient energy intake among Chinese adults in 2015–2017.

	18–49 y	50–64 y	65–79 y	≥80 y
Total	15,309 (60.5)	13,694 (57.8)	6594 (57.3)	849 (67.0)
Male	6881 (44.9)	6267 (45.8)	3930 (59.6)	495 (58.3)
Female	8428 (55.1)	7427 (54.2)	2664 (40.4)	354 (41.7)
*p*-values ^a^	<0.001	0.77	<0.001	<0.001

^a^ *p*-values are based on chi-square tests.

**Table 3 nutrients-16-02371-t003:** Differences in imbalanced nutrient intake of macro- and micronutrients by sex and age group.

	18–49 y (*n* = 25,295)			50–64 y (*n* = 23,697)			65–79 y (*n* = 11,509)			≥80 y (*n* = 1267)		
	Male (*n* = 11,035)	Female (*n* = 14,260)		Male (*n* = 10,864)	Female (*n* = 12,833)		Male (*n* = 5933)	Female (*n* = 5576)		Male (*n* = 674)	Female (*n* = 593)	
	*n* (%)	*n* (%)	*p* ^a^	*n* (%)	*n* (%)	*p* ^a^	*n* (%)	*n* (%)	*p* ^a^	*n* (%)	*n* (%)	*p* ^a^
Macronutrients(g) *												
Carbohydrates	203 (1.8)	530 (3.7)	<0.0001	190 (1.7)	449 (3.5)	<0.0001	144 (2.4)	297 (5.3)	<0.0001	30 (4.5)	61 (10.3)	<0.0001
RNI-Protein	6243 (56.6)	8689 (60.9)	<0.0001	6871 (63.2)	8230 (64.1)	0.1575	4296 (72.4)	4044 (72.5)	0.8888	535 (79.4)	463 (78.1)	0.5726
%E-Fat	5857 (53.1)	8302 (58.2)	<0.0001	6237 (57.4)	8066 (62.9)	<0.0001	3701 (62.4)	3728 (66.9)	<0.0001	427 (63.4)	386 (65.1)	0.5193
Dietary fiber ^d^	10,691 (96.9)	13,955 (97.9)	<0.0001	10,517 (96.8)	12,503 (97.4)	0.0042	5776 (97.4)	5492 (98.5)	<0.0001	663 (98.4)	585 (98.7)	0.6792
Vitamins												
A												
Insufficient intake ^b^	8683 (78.7)	10,706 (75.1)	<0.0001	8413 (77.4)	9537 (74.3)	<0.0001	4609 (77.7)	4211 (75.5)	0.0061	519 (77)	460 (77.6)	0.8095
Excessive intake ^c^	66 (0.6)	67 (0.5)	0.1619	69 (0.6)	55 (0.4)	0.0281	18 (0.3)	10 (0.2)	0.177	1 (0.1)	0 (0)	0.2611
B_1_ (Thiamin) ^e^												
Insufficient intake ^b^	9018 (81.7)	11,845 (83.1)	0.0053	9180 (84.5)	10,809 (84.2)	0.5671	5348 (90.1)	4958 (88.9)	0.0321	637 (94.5)	553 (93.3)	0.3505
B_2_ (Riboflavin) ^e^												
Insufficient intake ^b^	10,131 (91.8)	12,917 (90.6)	0.0007	10,105 (93.0)	11,682 (91.0)	<0.0001	5627 (94.8)	5192 (93.1)	<0.0001	638 (94.7)	550 (92.7)	0.1606
B_3_ (Niacin)												
Insufficient intake ^b^	3771 (34.2)	5074 (35.6)	0.0198	4182 (38.5)	4808 (37.5)	0.1041	2357 (39.7)	1986 (35.6)	<0.0001	323 (47.9)	182 (30.7)	<0.0001
Excessive intake ^c^	199 (1.8)	86 (0.6)	<0.0001	140 (1.3)	71 (0.6)	<0.0001	49 (0.8)	22 (0.4)	0.0031	6 (0.9)	5 (0.8)	0.9282
C												
Insufficient intake ^b^	7352 (66.6)	11,217 (78.7)	<0.0001	6906 (63.6)	9846 (76.7)	<0.0001	3898 (65.7)	4393 (78.8)	<0.0001	473 (70.2)	509 (85.8)	<0.0001
Minerals												
Calcium ^d^												
Insufficient intake ^b^	10,482 (95.0)	13,777 (96.6)	<0.0001	10,564 (97.2)	12,592 (98.1)	<0.0001	5779 (97.4)	5471 (98.1)	0.01	657 (97.5)	581 (98.0)	0.5537
Iron												
Insufficient intake ^b^	73 (0.7)	5059 (35.5)	<0.0001	73 (0.7)	309 (2.4)	<0.0001	101 (1.7)	275 (4.9)	<0.0001	28 (4.2)	61 (10.3)	<0.0001
Excessive intake ^c^	380 (3.4)	255 (1.8)	<0.0001	310 (2.9)	249 (1.9)	<0.0001	112 (1.9)	60 (1.1)	0.0003	3 (0.4)	4 (0.7)	0.5828
Sodium ^d^												
Insufficient intake ^b^	739 (6.7)	870 (6.1)	0.0541	652 (6.0)	648 (5.0)	0.0013	383 (6.5)	323 (5.8)	0.1387	47 (7.0)	58 (9.8)	0.0705
Excessive intake ^c^	10,072 (91.3)	12,991 (91.1)	0.632	9983 (91.9)	11,893 (92.7)	0.0239	5446 (91.8)	5112 (91.7)	0.8258	610 (90.5)	513 (86.5)	0.0254

Inadequate intake was defined as insufficient intake (for the macronutrients and vitamin B1 and B2) or, when an tolerable upper intake level (UL) was available, as insufficient or excessive intake (for all other vitamins and minerals). Insufficient intake: participants who had intakes below the estimated average requirement (EAR) or adequate intake (AI) for the nutrient of interest. Excessive intake: participants who had an intake that exceeded the UL for the nutrient of interest. * Participants who had intakes below the estimated average requirement (EAR) or adequate intake (AI) for macronutrients. ^a^ *p*-values are based on chi-square tests. ^b^ Insufficient intake was defined as intake of the nutrient of interest below the recommended EAR or AI. ^c^ Excess intake as intake of the nutrient of interest above the UL. ^d^ Adequate intake (AI) used as DRI instead of EAR. ^e^ No tolerable upper intake level (UL) has been established.

**Table 4 nutrients-16-02371-t004:** Association between demographics, BMI, nutritional status and inadequate micronutrient intake.

Age Groups	Multivariate Analysis						
		Male			Female		
18–49 y (*n* = 25,295)	Region; *n* (%)	Inadequate	OR (95%-CI)	*p*-values	Inadequate	OR (95%-CI)	*p*-values
	Eastern Region	2200 (34.4)	ref		3281 (34.9)	ref	
	Central Region	1794 (28.0)	1.4 (1.3,1.5)	<0.0001	2737 (29.1)	1.5 (1.3,1.6)	<0.0001
	Western Region	2411 (37.6)	1.5 (1.4,1.6)	<0.0001	3373 (36.0)	1.5 (1.3,1.6)	<0.0001
	Urban/rural; *n* (%)						
	Urban	2479 (38.7)	ref		3828 (40.8)	ref	
	rural	3926 (61.3)	1.1 (1.0,1.1)	0.163	5563 (59.2)	1.0 (0.9,1.0)	0.2498
	BMI categories; *n* (%)						
	Underweight	236 (3.7)	1.1 (0.9,1.4)	0.3851	413 (4.4)	1.0 (0.9,1.2)	0.6932
	Normal	2904 (45.3)	ref		4789 (51.0)	ref	
	Overweight	2268 (35.4)	1.1 (1.0,1.2)	0.1759	2923 (31.1)	1.0 (1.0,1.1)	0.232
	Obese	997 (15.6)	1.1 (1.0,1.3)	0.0274	1266 (13.5)	1.3 (1.1,1.4)	<0.0001
50–64 y (*n* = 23,697)	Region; *n* (%)						
	Eastern Region	2393 (37.2)	ref		3027 (37.7)	ref	
	Central Region	1975 (30.6)	1.3 (1.2,1.4)	<0.0001	2496 (31.1)	1.3 (1.2,1.4)	<0.0001
	Western Region	2070 (32.2)	1.5 (1.4,1.7)	<0.0001	2499 (31.2)	1.6 (1.4,1.7)	<0.0001
	Urban/rural; *n* (%)						
	Urban	2540 (39.5)	ref		3256 (40.6)	ref	
	rural	3898 (60.5)	1.0 (1.0,1.1)	0.4215	4766 (59.4)	1 (1.0,1.1)	0.3593
	BMI categories; *n* (%)						
	Underweight	165 (2.6)	1.2 (0.9,1.5)	0.2644	213 (2.7)	0.8 (0.7,1.0)	0.0457
	Normal	2939 (45.6)			3225 (40.2)		
	Overweight	2446 (38.0)	1.1 (1.0,1.2)	0.1778	3100 (38.6)	1.1 (1.0,1.2)	0.0339
	Obese	888 (13.8)	1.2 (1.1,1.4)	0.0021	1484 (18.5)	1.4 (1.2,1.5)	<0.0001
65–79 y (*n* = 11,509)	Region; *n* (%)						
	Eastern Region	1221 (35.6)	ref		1365 (37.2)	ref	
	Central Region	1101 (32.1)	1.4 (1.2,1.5)	<0.0001	1128 (30.7)	1.3 (1.1,1.5)	0.0001
	Western Region	1111 (32.3)	1.7 (1.5,1.9)	<0.0001	1178 (32.1)	1.5 (1.3,1.7)	<0.0001
	Urban/rural; *n* (%)						
	Urban	1457 (42.4)	ref		1680 (45.8)	ref	
	rural	1976 (57.6)	1.1 (1.0,1.2)	0.0654	1991 (54.2)	1.1 (0.9,1.2)	0.3802
	BMI categories; *n* (%)						
	Underweight	170 (5.0)	0.9 (0.7,1.2)	0.4263	208 (5.7)	1.3 (1.0,1.7)	0.0389
	Normal	1728 (50.3)	ref		1580 (43.0)	ref	
	Overweight	1188 (34.6)	1 (0.9,1.2)	0.5345	1318 (35.9)	1.2 (1.0,1.3)	0.0235
	Obese	347 (10.1)	1 (0.9,1.3)	0.637	565 (15.4)	1.1 (1.0,1.3)	0.1369
≥80 y (*n* = 1267)	Region; *n* (%)						
	Eastern Region	189 (40.6)	ref		194 (44.9)	ref	
	Central Region	138 (29.6)	1.8 (1.2,2.7)	0.005	127 (29.4)	1.7 (1.1,2.7)	0.0194
	Western Region	139 (29.8)	2.0 (1.3,3.1)	0.0012	111 (25.7)	1.5 (0.9,2.4)	0.1084
	Urban/rural; *n* (%)						
	Urban	198 (42.5)	ref		190 (44.0)	ref	
	rural	268 (57.5)	1.4 (1.0,1.9)	0.0747	242 (56.0)	1.3 (0.9,1.9)	0.1805
	BMI categories; *n* (%)						
	Underweight	39 (8.4)	1.4 (0.7,2.9)	0.3664	34 (7.9)	0.6 (0.3,1.2)	0.1697
	Normal	263 (56.4)	ref		261 (60.4)	ref	
	Overweight	128 (27.5)	0.8 (0.6,1.2)	0.3081	104 (24.1)	0.7 (0.4,1.1)	0.084
	Obese	36 (7.7)	0.9 (0.5,1.7)	0.723	33 (7.6)	0.4 (0.2,0.7)	0.0014

Inadequate micronutrient intake as <4 adequate micronutrient intakes (out of 8 in total). Reference category = inadequate micronutrient intake, defined as <4 adequate micronutrient intakes. NA = not applicable, REF = reference category, REF = reference category, OR = odds ratio. *p*-value = level of significance.

## Data Availability

The data presented in this study are non-public.

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
