# Peer review of "Deficiency of Energy and Nutrient and Gender Differences among Chinese Adults: China Nutrition and Health Survey (2015–2017)"

_nutrients, 2024, doi:10.3390/nu16142371_

Round 1

Reviewer 1 Report (Previous Reviewer 2)

Comments and Suggestions for Authors

nutrients-3079024_ Deficiency of Energy and Nutrient and gender differences among Chinese adults :China Nutrition and Health Surveillance (2015-2017)

This article is presented to the section “ Nutritional Epidemiology “ in the special issue “Dietary Patterns, Dietary Intake, Dietary Behaviours and Health  “.

This study utilized data from the China Nutrition and Health Surveillance (CNHS) collected between 2015 and 2017 to estimate the prevalence of inadequate dietary micronutrient intake among Chinese adults and to determine differences in micronutrient intake by gender, age, and BMI categories. A total of 61,768 subjects were included in this study, of which 33,262 (54%) were female.

The findings indicate that underweight and overweight women aged 65-79 were more likely to have inadequate micronutrient intake, whereas obese women over 80 years old were less likely to experience inadequate micronutrient intake. No significant differences in micronutrient intake were found between urban and rural areas for each age group.

Comments:

Given that the study focuses on micronutrients, the abstract should specify which micronutrients are being evaluated. As the abstract is a crucial section for attracting the interest of potential readers, this information should be clearly stated. Otherwise, the abstract is well-structured and effectively summarizes the study.

The introduction highlights the importance of the topic and utilizes relevant literature. However, it is recommended that the authors also address the significance of micronutrients for adult health.

Regarding the materials and methods section, it is described as a study based on a national survey. The strengths and weaknesses of this approach should be discussed in relation to micronutrient intake. The survey is family-based, followed by a 24-hour dietary recall for individuals, collecting data on 10 types of energy, 15 types of vitamins, 11 types of minerals, 20 types of amino acids, and 38 types of fatty acids. It is important to explain the rationale behind this selection.

Concerning the Recommended Dietary Allowances (RDA), since the study involves the Chinese population, it would be useful to explain any age or gender differences.

The presented tables in the results section are very informative.

The discussion is well-constructed, and the limitations are adequately addressed. However, considering that it is now 2024, the authors should consider potential changes in the nutritional characteristics of the population. The strengths of the study should also be included.

The conclusions are consistent with the results obtained.

Author Response

Given that the study focuses on micronutrients, the abstract should specify which micronutrients are being evaluated. As the abstract is a crucial section for attracting the interest of potential readers, this information should be clearly stated. Otherwise, the abstract is well-structured and effectively summarizes the study.

Response1:The results of the abstract show which micronutrients are deficient in Chinese adults, and accept your suggestion to focus on the micronutrients studied in this paper in the abstract method(line19)

The introduction highlights the importance of the topic and utilizes relevant literature. However, it is recommended that the authors also address the significance of micronutrients for adult health.

Response2:During the process of writing the thesis, I read a lot of reviews, especially the role of micronutrients in adult health, which was helpful for the discussion section to focus on the conclusions of this study and supplemented the literature to support the conclusions of this paper (line 251).

Regarding the materials and methods section, it is described as a study based on a national survey. The strengths and weaknesses of this approach should be discussed in relation to micronutrient intake. The survey is family-based, followed by a 24-hour dietary recall for individuals, collecting data on 10 types of energy, 15 types of vitamins, 11 types of minerals, 20 types of amino acids, and 38 types of fatty acids. It is important to explain the rationale behind this selection.

Response3:The average daily intake of energy, vitamins, and minerals per person was calculated based on the 2004 and 2009 editions of the Chinese Food Composition Tables. The nutrients included in the tables include 10 types of energy, water, ash, dietary fiber, and macronutrients, 15 types of vitamins, 11 types of minerals, 20 types of amino acids, and 38 types of fatty acids. This is the content of the book, and this study calculated the value of various nutrients based on the food intake data collected.

Reviewer 2 Report (New Reviewer)

Comments and Suggestions for Authors

Wei et al., Nutrients 2024

General:  Although the authors attempted to use the Nutrients template some of the details were not followed.  The maximum length of the abstract is 200 words. The authors will need to cut it by 40%.

Title: Gender, Differences, and Adults need to be capitalized.

Abstract:

L. 15-6: Fundamental flaw in the abstract:  The double burden is not overweight and obesity it is the overweight/obese and micronutrient deficiencies. This needs to be followed through the entire paper. The authors need to be clear on what the double burden of malnutrition is. 

Suggested deletions (among others): There are few studies on the comprehensive evaluation of micronutrients intake in Chinese adults. Inadequate energy intake occurs in adults of all ages.

L. 43:  Delete “material”

L. 44:  Delete “body”

L. 47: “marginal”

Introduction:

The entire second page is almost a single paragraph.  Please split to three paragraphs with common themes. For example, Lines 58snd 67 could start new paragraphs.

L. 55:  What does “its” refer to?

L. 71: The double burden is not overweight and obesity it is the overweight/obese and micronutrient deficiencies.

L. 76:  It is an estimate so use 790 million.

L. 79-80:  Retinol is vitamin A. Please revise.

L. 98:  “subjects” not “objects”

L. 97 and 100 are redundant.

L. 109:  Not a complete sentence.

L. 113:  Tense: “staff”

L. 114:  Delete “information”

L. 119:  “weighed” not “weighing”

L. 120:  Fix the spacing error.

L. 127: Tense: “recall”

Section 2.3 is not formatted well.  Please watch the spacing, especially around the parentheses. Why is “Reference” capitalized and not dietary or intake?  Be consistent.

L. 148:  “was” should be “were”

L. 160:  No need to capitalize “calcium” and “sodium”

L. 162:  “was” should be “were”

L. 181:  “lived”

Table 2 could be reformatted to be in portrait mode. 

Figure 2:  Should “Icon” be “Iron”?  Spell out “excess”

L. 207:  “micronutrient intakes”

Table 3:  Is cho, carbohydrate?

L. 237:  “micronutrients”

L. 239:  “low or high”?

L. 258-9:  Sentence is redundant and can be deleted.

L. 263 and 272:  Higher or lower? Do you mean deficiency?

L. 294-7:  Please use the appropriate punctuation.

Comments on the Quality of English Language

There are some tense issues.  A couple of areas need improvement. 

Author Response

General:  Although the authors attempted to use the Nutrients template some of the details were not followed.  The maximum length of the abstract is 200 words. The authors will need to cut it by 40%.

Response1:Thanks for your suggestion, I will cut down the characters in the abstract.

Title: Gender, Differences, and Adults need to be capitalized.

Response2:I have revised the title.

Abstract:

L.15-6: Fundamental flaw in the abstract:  The double burden is not overweight and obesity it is the overweight/obese and micronutrient deficiencies. This needs to be followed through the entire paper. The authors need to be clear on what the double burden of malnutrition is. 

Response3:I have revised it.

Suggested deletions (among others): There are few studies on the comprehensive evaluation of micronutrients intake in Chinese adults. Inadequate energy intake occurs in adults of all ages.

Response4:I have deleted it.

  1. 43:  Delete “material”L. 44:  Delete “body”L. 47: “marginal”

Response5:I have deleted.

Introduction:

The entire second page is almost a single paragraph.  Please split to three paragraphs with common themes. For example, Lines 58snd 67 could start new paragraphs.
Response6:I have modified it.

L.55:  What does “its” refer to?

Response7:Change its to their, referring to people with type 2 diabetes

L.71: The double burden is not overweight and obesity it is the overweight/obese and micronutrient deficiencies.

Response8:I have revised it.

  1. 76:  It is an estimate so use 790 million.

Response9:I have revised it.

  1. 79-80:  Retinol is vitamin A. Please revise.

Response10:I have revised it.

L.98:  “subjects” not “objects”

Response11:I have revised it.

  1. 97 and 100 are redundant.

Response12:I have deleted it.

  1. 109:  Not a complete sentence.

Response13:I have revised it.

  1. 113:  Tense: “staff”

Response14:This refers to the investigators involved in the inquiry

  1. 114:  Delete “information”

Response15:I have deleted it.

  1. 119:  “weighed” not “weighing”

Response16:I have revised it.

  1. 127: Tense: “recall”

Response17:I have revised it.

Section 2.3 is not formatted well.  Please watch the spacing, especially around the parentheses. Why is “Reference” capitalized and not dietary or intake?  Be consistent.

Response18:I have revised .

  1. 148:  “was” should be “were”L. 160:  No need to capitalize “calcium” and “sodium”L. 162:  “was” should be “were”  L. 181:  “lived”

Response19:I have revised .

Table 2 could be reformatted to be in portrait mode. Figure 2:  Should “Icon” be “Iron”?  Spell out “excess”L.207:  “micronutrient intakes”

Response20:I have revised .

Table 3:  Is cho, carbohydrate?

Response21:yes ,I have revised it.

L.237:  “micronutrients”

Response22:I have revised .

  1. 239:  “low or high”?

Response23:I should be”underweight’, i have revised it.

  1. 258-9:  Sentence is redundant and can be deleted.

Response24:I have deleted it.

L.263 and 272:  Higher or lower? Do you mean deficiency?

Response25:yes

L.294-7:  Please use the appropriate punctuation.

Response26:I have revised .

Comments on the Quality of English Language

There are some tense issues.  A couple of areas need improvement. 

Response27:Thank you for your hard work to review my article and put forward such sincere suggestions for revision. I would like to express my profound thanks here. I will revise it carefully, thank you.

This manuscript is a resubmission of an earlier submission. The following is a list of the peer review reports and author responses from that submission.

Round 1

Reviewer 1 Report

Comments and Suggestions for Authors

Thank you for the opportunity to conduct a review.

The topic discussed by the authors is very important, especially since it concerns one of the greatest social threats that knows no borders or statehood.

The authors presented the results in quite detail, but did not draw any scientific conclusions. Even a percentage comparison of the groups does not fully explain the situation and remains only a statistical comparison. There are no dependencies or attempts to explain the obtained results in the work. It's a waste of such large research material.

The study lacks information on how the content of micro and macro elements and vitamins in the consumed products was assessed. The authors indicated that the data were collected in the form of a 3-day nutritional questionnaire, but did not provide any information

1. who collected the data, whether they were recorded for the entire randomly selected household or for a single person

2. whether people participating in the project were instructed on how to evaluate individual portions

3. whether all studies were carried out in the same period (this especially applies to the consumption of fresh fruit and vegetables)

4. why the division of groups did not include information about education, family size, social status - this is very important information in lifestyle research, of which nutrition is one of the basis

5.when anthropometric measurements were performed

The work requires supplementing the methodology and results with the dependencies mentioned in the conclusions. The Discussion lacks an attempt to explain the observed differences - this is the most important thing in scientific work, why there are such differences

Author Response

Dear reviewer, thank you for taking your busy time to review the manuscript. Your questions are profound and meaningful. The following are my responses to the questions.

-The study lacks information on how the content of micro and macro elements and vitamins in the consumed products was assessed. The authors indicated that the data were collected in the form of a 3-day nutritional questionnaire, but did not provide any information

The calculation method used in this paper is introduced in the research method (see manuscript line125-132 for details). The per capita daily dietary intake of energy, vitamins and minerals is calculated according to the 2004 and 2009 editions of China Food Composition Table. The nutrients in the food composition list include 10 kinds of energy, water, ash, dietary fiber and macronutrients, 15 kinds of vitamins, 11 kinds of minerals, 20 kinds of amino acids, and 38 kinds of fatty acids.

1.who collected the data, whether they were recorded for the entire randomly selected household or for a single person

The Nutrition and Health Surveillance from 2015 to 2017 adopted the random cluster sampling method (see manuscript line98-113 for details) to investigate the personal information of sampled households; Investigators should be trained, have skilled professional skills and an honest attitude; After the assessment on the post. The investigation process is that the investigators go to the families of the respondents and conduct a face-to-face questionnaire survey, which is required to be completed within 15 to 40 minutes.

2.whether people participating in the project were instructed on how to evaluate individual portions.

Each investigator signed an informed consent form, and the investigator informed the respondents of the investigation project and results.

3.whether all studies were carried out in the same period (this especially applies to the consumption of fresh fruit and vegetables).

This project was conducted in 2015, and on-site surveys were carried out successively after training was conducted at 302 monitoring sites nationwide. The 24-hour dietary review method was generally conducted for 3 consecutive days, including 2 working days and 1 rest day, and households were surveyed on eating conditions 24 hours a day. Although this survey was not carried out at the same time, during the calculation, Weekday reuse has reduced the variation in food intake due to seasonal and holiday factors.
Moreover, the sample size of this study was large enough to provide a fairly reliable estimate of overall food consumption when the diet was relatively monotonous.

4.why the division of groups did not include information about education, family size, social status - this is very important information in lifestyle research, of which nutrition is one of the basis

This study focuses on two stratified variables, gender and age. Information about education, family size and social status will be studied in detail in the future.

Reviewer 2 Report

Comments and Suggestions for Authors

The article titled "Deficiency of Energy and Nutrient and Gender Differences among Chinese Adults: China Nutrition and Health Surveillance (2015-2017)" (nutrients-2883424) has been submitted to the "Nutritional Epidemiology" section in the Special Issue on "Dietary Patterns, Dietary Intake, Dietary Behaviours and Health."

This study assesses the deficiency rate of comprehensive micronutrient intake in the diets of Chinese adults, utilizing data from the China Nutrition and Health Surveillance (CNHS) spanning from 2015 to 2017. The analysis extends to exploring differences in micronutrient intake based on gender, age, and BMI. The findings serve as a crucial scientific foundation for policymakers to implement interventions aimed at addressing undernutrition and overnutrition in China.

The abstract is well-structured, accurately summarizing the content of the study. The introduction establishes the importance of the topic based on relevant literature.

In the methodology, it is advisable to specify that the study design is an ecological study with a population-based approach. If a different design is used, clarification is necessary. The description of the design should also be included in the abstract. Regarding data collection, the use of a 24-hour survey for three consecutive days is mentioned, which poses a significant limitation in assessing micronutrient intake due to their considerable variability. It is recommended to discuss the need for data collection over several days, depending on the type of micronutrient, in the discussion section.

The results are presented clearly through informative tables. The discussion is well-articulated, addressing gender and age group differences. However, strategies for addressing micronutrient deficiencies in the identified population are not proposed, and the limitations of the study design are not thoroughly discussed.

In conclusion, this study is valuable, providing relevant information for corrective measures in micronutrient intake. Nonetheless, it would benefit from suggesting potential strategies for addressing deficiencies and discussing the limitations of the study design. The findings also highlight the need for future research on this topic.

Author Response

Dear reviewer, thank you for taking your busy time to review the manuscript. Your questions are profound and meaningful. The following are my responses to the questions.
1. Add the limitations of dietary survey methods in the discussion section, especially the differences in seasonal foods;
2. Increase measures to improve micronutrient deficiency in special populations.

Round 2

Reviewer 1 Report

Comments and Suggestions for Authors

The authors have slightly supplemented the work, but it is still only a statistical review. There are no correlations in the results, and there is no attempt in the discussion to explain the results or compare them to other studies.

The answer to the question "were the people participating in the project instructed on how to assess individual portions" is unambiguous, I understand that each respondent signed a consent to the research - this is required by the Declaration of Helsinki, but there is no information whether each of the respondents knew what a spoon, slice, portion because this is how data is recorded in the nutritional questionnaire.

The authors wrote that the measurement of weight and body height was made by one researcher, so when this study was carried out with such extensive research in terms of group size and distance between individual areas covered by the project.

In my opinion, the work requires supplementing the statistics with correlations between the results obtained and age, gender, and place of residence.

Additionally, the work requires supplementing the discussion with comparable research

Author Response

1.The authors have slightly supplemented the work, but it is still only a statistical review. There are no correlations in the results, and there is no attempt in the discussion to explain the results or compare them to other studies.The answer to the question "were the people participating in the project instructed on how to assess individual portions" is unambiguous, I understand that each respondent signed a consent to the research - this is required by the Declaration of Helsinki, but there is no information whether each of the respondents knew what a spoon, slice, portion because this is how data is recorded in the nutritional questionnaire.

In the dietary survey, we used a dietary questionnaire validated by experts. Food measurement in China is different from that in foreign countries. We use a food scale to weigh the weight of household condiments, and the food we eat is asked to recall according to memory, in grams, and the raw weight and cooked weight are distinguished. However, in order to avoid the recall bias, we made a food map, and weighed some local food units (such as a biscuit, a cake, and a piece of bread) with a standard measuring plate and measuring bowl to help survey subjects accurately estimate the weight of the food.

2.The authors wrote that the measurement of weight and body height was made by one researcher, so when this study was carried out with such extensive research in terms of group size and distance between individual areas covered by the project.

There were 302 survey sites in this study, and a group of professionally trained investigators were selected for each survey site before conducting the survey. Therefore, there was not one researcher, which has been corrected in the manuscript.

3.In my opinion, the work requires supplementing the statistics with correlations between the results obtained and age, gender, and place of residence.Additionally, the work requires supplementing the discussion with comparable research

Age, gender and place of residence are demographic variables, which are used for group comparison and control of confounding factors in this study. This study is mainly to explore the relationship between gender and nutrient deficiency, and to explore the correlation between group variables is contrary to the content of this study and has no practical significance. In addition, in Part 3.5, the possibility of nutrient deficiency in adults of different genders has been analyzed at different ages, regions, urban and rural areas, and BMI status. This is one of the important conclusions of this study.

4.Additionally, the work requires supplementing the discussion with comparable research

 In the discussion of this paper, some research is added to supplement.
